# Relationship between Coronary Arterial Geometry and the Presence and Extend of Atherosclerotic Plaque Burden: A Review Discussing Methodology and Findings in the Era of Cardiac Computed Tomography Angiography

**DOI:** 10.3390/diagnostics12092178

**Published:** 2022-09-09

**Authors:** Georgios Rampidis, Vasileios Rafailidis, Konstantinos Kouskouras, Andjoli Davidhi, Angeliki Papachristodoulou, Athanasios Samaras, George Giannakoulas, Antonios Ziakas, Panagiotis Prassopoulos, Haralambos Karvounis

**Affiliations:** 1First Department of Cardiology, AHEPA University Hospital, Aristotle University of Thessaloniki, 54636 Thessaloniki, Greece; 2Department of Radiology, AHEPA University Hospital, Aristotle University of Thessaloniki, 54636 Thessaloniki, Greece

**Keywords:** coronary, atherosclerosis, cardiac CT, geometry

## Abstract

Coronary artery disease (CAD) represents a modern pandemic associated with significant morbidity and mortality. The multi-faceted pathogenesis of this entity has long been investigated, highlighting the contribution of systemic factors such as hyperlipidemia and hypertension. Nevertheless, recent research has drawn attention to the importance of geometrical features of coronary vasculature on the complexity and vulnerability of coronary atherosclerosis. Various parameters have been investigated so far, including vessel-length, coronary artery volume index, cross-sectional area, curvature, and tortuosity, using primarily invasive coronary angiography (ICA) and recently non-invasive cardiac computed tomography angiography (CCTA). It is clear that there is correlation between geometrical parameters and both the haemodynamic alterations augmenting the atherosclerosis-prone environment and the extent of plaque burden. The purpose of this review is to discuss the currently available literature regarding this issue and propose a potential non-invasive imaging biomarker, the geometric risk score, which could be of importance to allow the early detection of individuals at increased risk of developing CAD.

## 1. Introduction

Cardiovascular disease, including CAD, constitutes the first cause of mortality in developed countries and particularly societies with a western lifestyle. Reports show that approximately one third of all deaths in patients older than 35 years are attributed to chronic heart disease. It is also estimated that 4 million patients die in Europe every year, while in Greece CAD is characterized by a mortality of 110 deaths per 100,000 people and 16,000 new cases of stable angina present every year [1,2]. These facts show that coronary atherosclerotic disease is a very important modern pandemic with multi-faceted consequences for both the patient and the healthcare system. 

CAD is primarily associated with the formation of atherosclerotic plaques along the course of coronary arteries, a phenomenon not yet fully understood but with many contributing pathophysiologic factors. The currently accepted mechanism for coronary atherosclerosis includes the following stages: an endothelial malfunction and the subendothelial deposition of low-density lipoprotein (LDL); the oxidization of LDL; the migration of monocytes towards the subendothelial level and their transformation to macrophages; foam-cell formation; the multiplication of smooth muscle cells; and finally the apoptosis of foam cells, leading to the creation of necrotic cores. There are many systematic factors contributing to this multi-faceted process of plaque development, including hyperlipidemia, hypertension, genetic factors, and vessel characteristics defining the wall shear stress and other blood flow hemodynamic factors [3,4]. 

The notion that vascular geometric characteristics influence atherosclerosis development and progression is not new but has still not been fully studied so far due to the complexity of the issue and the abundance of different vascular visualization techniques. The predisposing factors of atherosclerosis mentioned previously apply everywhere within the vascular network of a patient and would thus be expected to lead to a diffuse and uniform thickening of the vascular wall. Nevertheless, it has been long known that different vessels or segments of the same vessel are affected to a different degree by atherosclerotic plaque formation. Namely, atherosclerotic plaques tend to occur near the origins of arterial branches or in vascular bends, due to specific flow patterns observed in such vascular segments. Such geometric parameters create an atherosclerotic-prone environment favoring either high or low shear stress [5]. Interestingly, it was found that atherosclerotic plaques typically affect the outer walls of left coronary bifurcation, while the flow divider and the inner walls downstream were relatively free of disease [6]. Hence, it is possible that vascular inherent characteristics would either favor or prevent the development of atherosclerosis. For instance, studies have shown that atherosclerosis is common in the medial aspect of curved vessels or lateral walls of bifurcations [7,8]. An example is represented by the localization of atherosclerotic plaques on the inner curvature of the aortic arch and close to arterial branches orifices, albeit the uniform influence of systemic factors such as hypertension, hypercholesterolemia, and diabetes [4]. It is plausible that the influence of vascular geometry on atherosclerosis is expressed through parameters describing the effect of blood on the vascular wall, such as the tensile stress (TS) and the shear stress (SS) [9].

Modalities traditionally used for the evaluation of CAD, including invasive coronary angiography (ICA) and ultrasound, are both limited by their two-dimensional nature, while atherosclerotic vessels represent a three-dimensional structure. The introduction of modern scanners and technologies rendered coronary computed tomography angiography (CCTA) a superb and widely available modality for the evaluation of coronary anatomy and disease [10]. This technique provides us with the possibility to readily assess and quantify coronary arterial geometry and location and extent, and the grade of atherosclerosis. Nonetheless, questions still remain regarding the optimal parameter with which to quantify coronary arterial geometry, and its reproducibility, availability, and applicability in everyday clinical settings, rather than regarding the scientific research setting or the clinical significance of each parameter. The purpose of this review is to discuss the literature regarding the influence of coronary arterial geometry on atherosclerosis development (Table 1) and suggest a potential tool for the quantification of geometric complexity.

## 2. Technical Considerations on the Description of Coronary Arteries Geometry

As it was previously suggested, the relationship between vascular geometry and atherosclerosis could be mediated by analyzing the influence of geometry on hemodynamics and flow profile. Nevertheless, in the era of modern imaging, the geometry can be directly correlated with the extent of atherosclerosis using CCTA, and this is the current target of ongoing research [11]. The coronary geometry can be described using a multitude of parameters, which will be described in this section. Importantly, whichever geometric assessment is to be used, it needs to be relatively easy and not time-consuming, in order to be able to be readily incorporated into the everyday clinical practice and to be widely accepted by CCTA reporting physicians and clinicians.

In an angiographic study of autopsy hearts, Brinkman et al. attempted to define the variability of the coronary tree by using specific geometric parameters [12]. This study used pairs of projection angiograms and focused on the left anterior descending artery (LAD) and its first two major branches. The parameters studied included (i) the angle between LAD and the left circumflex (LCx), (ii) the angles between LAD and the early diagonal and septal perforator branches, (iii) the distance between branch points, and (iv) tortuosity. A considerable variation of these geometric features was noted, with most parameters showing no correlation. Only the angle between LAD and the second diagonal branch was significantly correlated with the distance between the ostia of the two first diagonal arteries. Interestingly, no correlation was found between the geometric variables and demographics, including age and gender. Nonetheless, species from black patients exhibited a larger LMA angle than whites [12]. This study showed that the left main artery (LM) angle had a mean value of 76.4° with an SD of 16.7°. The means and SD for the first and second diagonal branch angle measured 47.3 ± 14.6° and 48 ± 16.8°, respectively. These data suggested that the existing broad variability of the coronary tree is a factor potentially causing haemodynamic alterations. Brinkman et al. suggested that the first two major branches of the LAD are defined as those whose diameter is at least 2 mm or 1 mm. This can be used as a suggestion as to which vessels should be included in a study of geometric complexity or the generation of a geometric risk score [12]. In this study, the following geometric features were defined and used: assuming two defined points along the course of a vessel (for instance the ostia of two branching vessels), the straight line connecting those points can be defined as straight distance (D_s_), while the contour distance reflected the actual length of a vessel following a rounded course (termed as Dc). The tortuosity was defined as (D_c_/D_s_) − 1, so that a straight vessel would yield a value of 0 tortuosity. In this study, the curvature was determined from vector representation of the vessel axis [12].

The variation of the anatomy of the coronary tree was further studied and catalogued in living healthy subjects during clinical biplane cine-angiography by Zhu et al. in 2009 [13]. This group measured and reported in detail the geometric variables of both the left and right coronary circulation and all proximal, middle, and distal segments of the vessels, including curvature, torsion, and tortuosity. Analysis showed that the LAD exhibits the highest curvature, torsion, and tortuosity in its distal portion, while in the right coronary artery (RCA), the same variables are lowest in its middle segment. Moreover, the LAD yielded significantly higher torsion value than the RCA [13]. The value of this study was that it catalogued the normal geometric features of both coronary arteries in end-diastole so that future studies could use this detailed registry as a reference to identify risk factors for atherosclerosis [13]. Gauss et al. also provided a descriptive analysis of coronary geometric features using CCTA [14]. In this study, the LM bifurcated in 87% of patients, trifurcated in 12%, and quadrifurcated in 1%. Features described in this study include the length of the LM and the average bifurcation angles in both systole and diastole. Interestingly, all angles measured were significantly larger in the systole. It was also noted that coronary systems with trifurcation configuration were characterized by significantly wider angles and longer LM segments, compared to systems with bifurcation [14].

One essential confounding factor for the study of coronary arterial tree anatomy and geometry is the fact that coronary arteries exhibit a significant movement throughout the cardiac cycle. This can be studied using clinical biplane cine-angiograms, and data so far show that both the LAD and RCA show significant variation throughout the cycle for the measures of curvature and torsion. This continuous motion is thought to influence both vessel wall mechanics and fluid dynamics, thus having implications on atherosclerosis development. Significant coefficients of variation were noted using in vivo tracking of the coronary arteries motion, with individual vessels coefficients reaching 2.28. It was found that the LAD shows greater variability of its torsion, compared with the RCA but less displacement [15]. The feature of torsion was described in this study as a parameter that reflects the degree to which a curved line (or vessel) remains within one plane or moves towards a certain direction, thus being part of a helix and showing a constant torsion along its length [15].

The observation that dynamic changes in coronary geometry throughout the cardiac cycle influence the atherosclerosis onset and progress was also confirmed by Zhu et al. in 2003. This study used biplane cine-angiograms to assess the right coronary artery, measuring vessel length, cyclic displacement, axial strain, curvature, and torsion. Intravascular ultrasound (IVUS) was performed to measure the vascular wall thickness. Although independent geometry parameters failed to correlate with wall thickness, the linear combinations of parameters did predict thickness with high confidence (R^2^ between 0.17 and 0.44). Both of the time-average values of curvature and torsion and their excursion throughout the cardiac cycle were positively correlated with maximum wall thickness and cross-sectional asymmetry [16]. The cardiac cycle effects on coronary geometry were further studied by Katranas et al. using in vivo CCTA to produce models for computational fluid dynamics (CFD) analysis [17]. This study confirmed the changes occurring during the cardiac cycle, showing that molecular viscosity was higher in systole, whereas endothelial shear stress was lower. The latter applied both in normal and atheromatic vessels. The feature of curvature was higher in systole and in atherosclerotic segments [17]. In a different setting, the dynamic changes throughout the cardiac cycle were also demonstrated using cardiac MRI in a healthy volunteer. MRI proved able to record the expected variation of vessel radius and curvature during the various phases of the cardiac cycle, while the dynamic motion of RCA had a significant impact on endothelial shear stress (ESS) calculated by CFD analysis [18].

In contrast to previous studies, van Zandwijk et al. concluded that there was no significant association between the degree of stenosis and plaque types and the dynamic changes observed in geometrical parameters during the cardiac cycle [19]. Similar to previous study designs, this research group used CCTA with retrospective electrocardiographic gating to acquire end-systolic and end-diastolic phase reconstructions of the coronary vessels and measured curvature and tortuosity for both phases. It was found that curvature was significantly higher in end-systole for both artery and segment level. Tortuosity, on the other hand, was significantly different only at a segment level [19]. Considering the dynamic changes of geometry throughout the cardiac cycle is expected to add to the study of atherosclerosis development but at the same time introduces more complexity and workload.

## 3. Association of Coronary Geometry with Haemodynamic Alterations

Although the natural history of atherosclerosis is a complex and dynamic process determined by a variety of local and systemic biological and biomechanical factors, blood-flow-induced shear stress has emerged as an essential feature of atherogenesis. ESS is the tangential force exerted on the endothelial surface that results from the friction of the flowing blood. The pattern of blood flow through a vessel depends on the flow velocity and the presence of geometric irregularities or obstructions. Flow may be either laminar or turbulent [Figure 1]. In relatively straight segments, ESS is pulsatile and unidirectional, with a magnitude varying between 15 and 70 dynes/cm^2^ over the cardiac cycle, and is known to be crucial for normal vascular functioning and atheroprotection [20,21]. In contrast, in geometrically irregular regions, as coronary bifurcations, turbulent flow occurs, and pulsatile flow generates low and/or oscillatory ESS. Low ESS refers to ESS, which is unidirectional but has a periodically fluctuating magnitude that results in a significantly low time-average (approximately less than 15 dyne/cm^2^). Low ESS typically occurs at the inner areas of curvatures and upstream of stenoses (Figure 1). Oscillatory ESS is characterized by significant changes in both direction (bidirectional) and magnitude over the cardiac cycle, resulting in a very low time-average, usually close to zero. Oscillatory ESS occurs primarily downstream of stenoses, at the lateral walls of bifurcations and in the vicinity of branch points. Low and oscillatory ESS promote a mechanical pro-atherogenic environment that leads to vascular dysfunction and atherosclerosis progression [3,7,21]. On the other hand, high ESS (more than 70 dynes/cm^2^) may occur concurrently at the “neck” of a stenosis, increasing in magnitude with progressive luminal encroachment. This enhances local thrombogenicity and triggers molecular pathways implicated in fibrous cap disruption, increasing the probability of clinical manifestation as acute coronary syndrome [21]. It has been proposed that in analogy to the Virchow’s triad, the development of atherosclerosis on the arterial wall is governed by (i) haemodynamic factors, including flow and vascular wall geometric features; (ii) the biological response of the arterial wall on a cellular level; and (iii) systemic risk factors).

An experimental study published in 1983 analyzed haemodynamic data in pulsatile flow through human aortic bifurcation casts [5]. In this study, laser Doppler anemometry was used to measure flow velocity at multiple sites within the vessel cast. Four geometric features have been determined, showing variability between different individuals, a phenomenon potentially explaining variability in the location and rate of development of atherosclerosis. The geometric features were (i) the presence of a flow divider; (ii) the focal curvatures of the vessel; (iii) the origin of a branch vessel with an unusually large angle, leading to inordinately low shear stress applied to the outer wall of this branch; (iv) bifurcation, where one branch exhibits a large angle with the parent vessel and one a small one, with the latter being exposed to relatively low ESS. The variability of these features among different individuals was proposed as a potential source of variability of atherosclerosis, and the term “geometric risk factors” was suggested [5].

The relationship between the geometric features of coronary bifurcations and local haemodynamic parameters was studied in models of both healthy and stenotic vessels using CFD simulations. The study by Chiastra et al. focused on the LAD/diagonal angle and showed that the bifurcation angle mildly influences the hemodynamics in both stenosed and healthy vessels [20]. On the contrary, the curvature radius was found to influence the generation of helical flow and helicity intensity, with an inverse correlation in both healthy and stenotic models. This effect was further magnified in stenotic vessels. Curvature was also found to influence the near wall haemodynamics of stenotic vessels, with a smaller value of curvature resulting in higher exposure to low ESS and lower exposure to oscillatory ESS. In this study, the curvature of a coronary bifurcation was defined as the radius (R) of a sphere on which the bifurcation lies and consequently bends, meaning that the lower the R, the more “curved” the bifurcation. A value of 56.3 mm was considered to be a physiological curvature in this study, whereas extreme values of ∞ and 16.5 mm were also considered [20]. Another study with CFD showed that wider angles of the left coronary system are associated with a disturbed flow pattern and low ESS gradient in bifurcations [22].

Among its multiple benefits, modern CCTA can help the investigation of the influence of coronary geometry on atherosclerosis by providing accurate data for the production of models in order to perform CFD simulation using patient-specific models. Such a study was performed by Peng et al. [23]. This study found that degree of stenosis is positively and significantly associated with maximum ESS at the intra-stenotic region. On the contrary, the recirculation zone length is moderately associated with the curvature and length of the lesion segment. The bifurcation of coronary arteries was significantly correlated with the occurrence of recirculation, while the distal stenosis severity exhibited an effect on the alteration of flow in the upstream bifurcation [23]. This study was in keeping with previous observations that there are clearly defined areas of the vascular wall that are exposed to high and low ESS, found in vascular bifurcations [24,25], areas prone to atherosclerosis formation. Another study using in vivo assessment of coronary arteries with CT and CFD showed that the tortuosity index of the left main and LAD segment influenced the low ESS observed in the proximal segment of the LAD, highlighting the emerging role of this modality in the evaluation of coronary geometry and its implications on atherosclerosis development [26].

The association between left coronary artery geometry and haemodynamic characteristics was addressed by Pinho et al., who used CTA data to perform CFD analysis of the left coronary circulation. This research group evaluated the vessel’s cross-section area; the proximal LAD length; and the angles between branches and the septum, curvature, and tortuosity. It was concluded that arteries with high caliber, high angles (LMS-LAD, LAD-LCx and LAD-septum), and high tortuosity were significantly correlated with a haemodynamic profile favoring plaque formation within the LAD. In contrast to these findings, increased proximal LAD length, angle (LMS-LCx), tortuosity (LMS-LAD and LAD-LCx), and curvature of LMS and LCx lead to a more atherosclerosis-resistant haemodynamic profile [27]. A similar study design for the right coronary circulation was also published by Pinho et al., reaching comparable findings [28]. Namely, it was shown that the association of geometry and haemodynamic disturbances was stronger in the middle and distal segment of the RCA. The ostium area of the RCA was positively correlated with the ESS magnitude from the proximal to distal segment of the vessel. The characteristics of the right ventricular branch, including the cross-sectional area, angle, and tortuosity, were strongly positively correlated with the ESS magnitude in the middle and distal segments of the vessel [28].

## 4. Association of Coronary Geometry with Atherosclerosis Lesions and Indexes

The theory that arterial geometry affects atherosclerosis onset and formation is not new, with reports on pulsatile flow through vascular branches and the hemodynamics of atherogenesis dating back to 1975 [29]. In a study published in 1976 by Gazetopoulos et al., it was shown that the length of the main left coronary artery was significantly shorter in patients with coronary atherosclerosis, as demonstrated on invasive angiography [30]. This finding was confirmed by Saltissi et al., who showed that the LM was shorter in atherosclerotic vessels, as documented in ICA. In particular, it was suggested that a short left main favored proximal atherosclerosis [31]. However, the results from both studies are limited by the small sample size and the single center nature.

In an early study published in 1993, Friedman et al. investigated the correlation of coronary artery geometry with the distribution of early atherosclerotic lesions. The angle formed between LCx and LAD arteries in the bifurcation of LMA was measured based on the multi-plane angiograms of the left coronary artery in fifteen autopsy specimens. The images were digitized and computationally processed in order to generate three-dimensional models of the centerlines of the lumens. A normalized index of disease severity RPI (relative proximal involvement) was used to isolate the effect of geometry on disease development (on the first segment of a vessel, adjacent to the bifurcation) from other risk factors that would apply to the entire vessel uniformly. This index divided the percentage of atherosclerotic plaque burden seen in the first 1 cm post the bifurcation by the percentage seen in the first 5 cm. Interestingly, no correlation was found between the angle and the extent of atherosclerotic burden in either the first 1 or 5 cm of the vessels. This is why the authors used the RPI, in an attempt to eliminate the effect of systemic factors, applying to the entire vessel uniformly and only study the effect of geometry on the proximal segment of the vessel. It was concluded that the RPI was negatively correlated with the LM branch angle. As a result, a small angle would lead to proximal atherosclerotic disease, an observation stronger for LCX. This observation was in-keeping with the notion that atherosclerosis affects vessels with values of ESS near zero. The stronger correlation observed with LCX was attributed to the fact that LAD is usually a continuation with LM, while LCX is usually a side-branch, originating with a larger angle as measured by ICA [32]. Contrary to this study, Saltissi et al. found no significant difference in the left main angle between healthy and atherosclerotic vessels on invasive angiography, although the authors noted a tendency for a wider bifurcation angle in patients with proximal disease. These authors combined the length of LM and LM bifurcation angle, finding that the combination of a short left main with a wide bifurcation angle was associated with proximal localization of atherosclerotic plaques, to a higher degree that a short LM alone did. No correlation was found between dominance of the LCx and localization of atherosclerotic plaques [31].

A multi-plane angiographic study performed on autopsy hearts, with histologic examinations of the vascular wall, investigated the relationship between the angle of origin of LAD branches and the structural asymmetry of the branch vessels. It was found that the angle of the branch origin was positively correlated with the maximum thickness of intima and media, as well as the branch’s circumferential asymmetry. This shows that branches originating with a larger angle from the parent vessel show eccentric intimal thickening and are thus prone to atherosclerosis formation [33]. An attempt was made to examine the effect of geometry on vascular wall morphology, even before the occurrence of atherosclerosis, with autopsy heart of lesion-free individuals examined with angiogram and histology. Even in this stage of absence of disease, it was found that the angle of an LAD branch proximal to the site under examination positively correlated with intimal and medial thickness, while the local curvature was also correlated with the maximum thickness of the intima and media layer. These observations establish a relationship between the geometry and the morphometric characteristics of coronary arteries, even in patients with no coronary atherosclerotic disease [34].

In a recent study, CCTA was used to quantify coronary geometry and correlate it with atherosclerotic burden [35]. Coronary geometry was quantified by means of curvature and tortuosity using the vessels centerline in asymptomatic patients. Curvature was significantly associated with significant stenosis both at a per-segment and per-artery level, being 16.7% higher in segments with stenosis and 13.8% higher in arteries with stenosis. On the other hand, tortuosity was associated with significant stenosis only at the per-segment level. The presence of plaque was related to curvature both on a segment and artery level, whereas it was related to tortuosity only on a segment level. In this study, end-diastolic phase reconstructions were used, proving that even static analysis can provide meaningful information, albeit previous findings favoring dynamic analysis of the cardiac cycle. This study showed that CCTA can successfully produce geometric features associated with the occurrence of atherosclerosis, highlighting the added value of this imaging modality in the field of cardiac disease [35].

Altintas et al. examined a patient population with ST elevation myocardial infarction of the RCA (*n* = 163), using ICA to classify the RCA shape as C-shaped or S-shaped based on the left oblique position angiographic view [36]. It was found that C-shaped RCAs (*n* = 124) were significantly more affected by atherosclerotic lesions in their proximal and mid regions. The TIMI frame count (TFC) was significantly higher in S-shaped RCAs [36].

The association between lesion-level adverse geometric characteristics (AGCs) and the risk of future acute coronary syndrome (ACS) was addressed by Han et al. [37], who used non-invasive cardiac imaging in a multicenter nested case-control cohort study, including patients with ACS and a culprit lesion precursor identified on baseline CCTA (*n* = 116) and propensity score-matched non-ACS controls (*n* = 116). Atherosclerotic plaques a short distance from the coronary ostium or located within vessel bifurcations or tortuous segments were more likely to develop into culprit lesions causing an ACS. The research group concluded that CCTA-derived AGCs capturing lesion location and vessel geometry are associated with the risk of future ACS-causing culprit lesions. In particular, it was suggested that adverse geometric features may provide additive prognostic information beyond plaque assessment in CCTA.

In another study published in 2020, Benetos et al. investigated the long-term prognostic value of coronary artery volume per myocardial mass as a potential new imaging biomarker [38]. To study this, 325 patients were followed for 4.6 years, with 11.1% cardiovascular events. Patients with a low artery volume/myocardial mass ratio (<28 mm^3^/g) had four times more events than patients with high coronary artery volume index (CAVi), regardless of the presence of significant stenoses. Such a result supports the idea of a normalization of atheroma burden to the volume of the coronary arterial tree. Moreover, the same research group demonstrated that CAVi is independently associated with abnormal stress myocardial blood flow (MBF), as derived from ^13^N-ammonia positron emission tomography myocardial perfusion imaging [39]. More importantly, CAVi exhibits incremental value to predict abnormal MBF and ischemia over stenosis severity from CCTA alone.

Finally, the ongoing GEOMETRY-CTA study [40] aims to introduce a quantitative, non-invasive imaging biomarker utilizing coronary artery geometric features, clinical and genetic risk factors, serum biomarkers, epicardial fat volume, and the coronary artery calcium score (CACS), to predict the presence and the complexity of CAD on CCTA and investigate its prognostic value regarding adverse cardiovascular events (Figure 2). The study intends to recruit 100 consecutive patients with suspected CAD and low/intermediate pre-test probability. Coronary geometrical characteristics such as the angulation of coronary bifurcations, tortuosity, CAVi [38], and vessel-length will be assessed with multi-planar reformation and volume rendering techniques (Figure 3) and integrated into a single geometric risk score. The extent and vulnerability of plaque burden will be calculated using several anatomical scoring systems such as the Leiden CTA risk score [41] and CT-adapted Gensini score [42]. Patients will be prospectively followed for 12 months after enrollment. In clinical practice, the utilization of such an approach could improve risk stratification and help guide downstream personalized management. Furthermore, the derived index will be available for incorporation in larger national prospective studies for further cardiovascular risk stratification.

**Table 1 diagnostics-12-02178-t001:** Relationship between main geometric features proposed in the literature and atherosclerosis.

Parameter	Outcome	Modality	Number of Patients [*n*]
**Length of LM** [30,31]	-Shorter in CAD patients	-ICA	43 [30] 149 [31]
**Angle LCX-LAD** [32]	-Negative correlation with CAD in proximal segments of LCX and LAD.	-angiography in autopsy hearts	15
**LM length and angle** [31]	-A short LM with wider angles was associated with proximal stenosis	-ICA	149
**Angle LAD-branch** [33]	-Positive correlation with maximum IMT and circumferential asymmetry	-angiography in autopsy hearts	14
**Angle LAD-Diagonal** [20]	-Mild influence on hemodynamics	-CFD/models	10
**Curvature radius** [20]	-Generation of helical flow	-CFD/models	10
**Degree of stenosis** [23]	-Positive correlation with maximum wall shear stress at the point of stenosis	-CFD/CCTA	22
**Tortuosity and curvature** [35]	-Significant positive correlation with significant stenosis and plaque formation	-CCTA	73
**Left coronary artery** **Cross-section area, angles (LM-LAD, LAD-LCx, and LAD-septum), and tortuosity** [27]	-Association with atherosclerosis prone haemodynamic profile	-CFD/CCTA	8
**Right coronary artery** **ostium area and RV branch cross-sectional area, angle and tortuosity** [28]	-Positive correlation with wall shear stress magnitude from the proximal to the distal segment	-CFD/CCTA	10
**Shape of the RCA** C **or S shaped** [36]	-TFC higher in S-RCAs -C-RCAs were significantly more affected in proximal and mid segments	-ICA/STEMI	163
**CAVi** [38,39]	-Lower CAVi (<28 mm^3^/g) × 4 more CV events -Lower CAVi associated with abnormal stress MBF and ischemia	-CCTA/PET	325 [38] 60 [39]

CAD, coronary artery disease; ICA, invasive coronary angiography; LAD, left anterior descending; RCA, right coronary artery; LM, left main; LCX, left circumflex; IMT, intima-media thickness; CFD, computational fluid dynamics; CCTA, cardiac computed tomography angiography; RV, right ventricular; TFC, TIMI frame count; STEMI, ST elevation myocardial infarction; CAVi, coronary artery volume index; CV, cardiovascular; MBF, myocardial blood flow; and PET, positron emission tomography.

## 5. Conclusions

Coronary atherosclerotic disease represents an undeniable modern pandemic with important consequences in terms of morbidity and mortality, particularly in western societies. The pathogenesis of this disease has long been studied, with many factors identified as contributing to its multi-faceted nature. Such traditional factors include systemic conditions such as hyperlipidemia and hypertension, the effect of which on the vascular wall is expected to be uniform throughout the vascular tree. Nevertheless, the observation that atherosclerotic plaques tend to affect particular points of the vascular tree raised the effect of local geometric features to the occurrence and progression of atherosclerosis. The relationship between the coronary arterial tree geometry and atherosclerotic disease has gained interest over the past years, with several reports investigating multiple aspects of this interaction. It is clear that coronary geometry influences atherosclerosis, but several aspects still require definitive answers. Such pending questions include which modality should be used (ICA vs. CCTA), whether dynamic or static measurements should be made, which exact parameters among the numerous reported should be used, and to which exact points of the coronary tree the measurements should be performed. It is widely recognized that state-of-the-art CCTA, due to the three-dimensional imaging nature, offers an unparalleled technique for the assessment of coronary arterial geometry and is expected to further advance knowledge in this field.

## Figures and Tables

**Figure 1 diagnostics-12-02178-f001:**
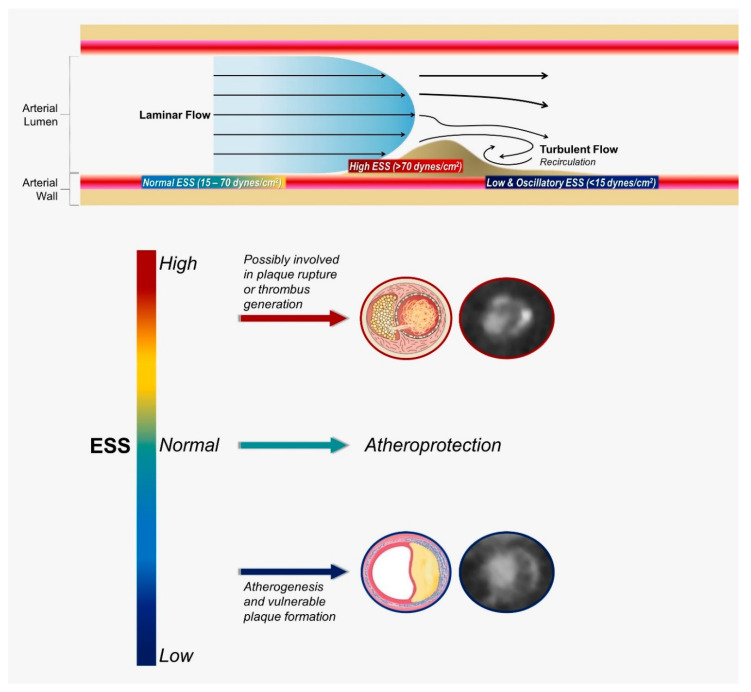
Blood flow patterns, levels of shear stress, and their effects on atherosclerosis. ESS, endothelial shear stress.

**Figure 2 diagnostics-12-02178-f002:**
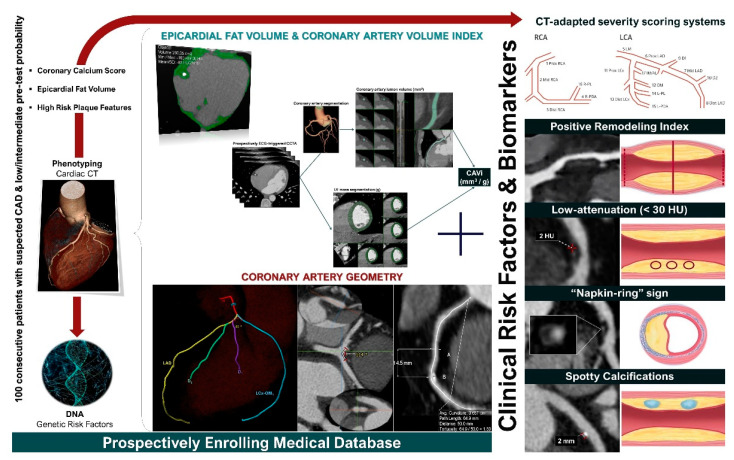
GEOMETRY-CTA study design (ClinicalTrials.gov ID: NCT04185493).

**Figure 3 diagnostics-12-02178-f003:**
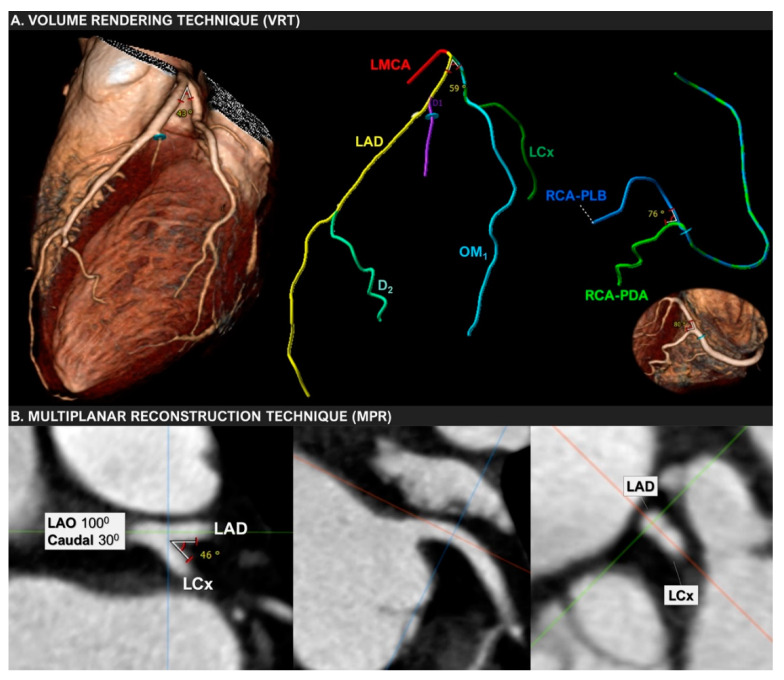
Volume-rendering (VRT) and multiplanar reconstruction (MPR) techniques used to measure the bifurcation angle between LAD and LCx in a 67-year-old woman with mild non-obstructive coronary artery disease.

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
