# Peer review of "Relationship between Coronary Arterial Geometry and the Presence and Extend of Atherosclerotic Plaque Burden: A Review Discussing Methodology and Findings in the Era of Cardiac Computed Tomography Angiography"

_diagnostics, 2022, doi:10.3390/diagnostics12092178_

Round 1

Reviewer 1 Report

In this review article, the authors discussed about the currently available literature regarding issues and proposed a potential non-invasive imaging biomarker, the geometric risk score, which could be of importance to allow early detection of individuals at increased risk of developing CAD.

1.       Abstract seems to describe just backgrounds and purpose. It is essential to summarize whole manuscript in the abstract appropriately for all the readers' convenience.

2.       Page 2, line 58. The authors used the words ‘unusual flow’, and described the flow dynamics in this review articles. I recommend the authors 1) to provide figures which describe flow pattern (laminar, turbulent, recirculation, high WSS, low WSS) to let readers understand it in detail at a glance, and 2) to explain basic fluid dynamics theory. Except for the basic knowledge and understanding of fluid dynamics, it would be difficult to understand this review article.

3.       Figure 1. It seems difficult to understand meanings of this figure to only see the figure and figure legends. The authors should include more information in the figure legends, and add some panels which describe the CFD results of WSS, stream line or OSI.  

4.       Page 6, line 193. It should not be concluded that high WSS is athero-protective widely. Both low WSS and high WSS were reported to be associated with atherosclerosis (Circulation. 2011 Aug 16;124(7):779-88. doi: 10.1161/CIRCULATIONAHA.111.021824.) This association should be the relation ‘Which came first, the chicken or the egg?’. In my opinion, moderate WSS might be protective for atherosclerosis, and abnormal (lower and higher) WSS would have a role in the progression of vulnerable plaque.

5.       Page6, line 191. The authors used the words ‘oscillatory wall shear stress’ 3 times. This concept is difficult to understand without explanations. Some description about the basic knowledge of oscillatory wall shear stress should be included.

6.       There are no graphic abstract data in the review site, I recommend to make a figure which make it easy to understand the summary of this review article.

Reviewer 2 Report

This is a well-written review for coronary artery geometry and atherosclerosis.

There are minor comments for authors:

Line 34, 35: use commas for numbers.

L 70: make a new paragraph here.

Table 1: some items and outcomes are difficult to follow. Please state clearly. It may be better to select items with positive correlations or to provide a table for high-risk geometric items.

L 146, 159: use small letters for Cine.

L 274: add (RPI).

L 300: please define "large" and "narrow" angles.

Reviewer 3 Report

The review covers and interesting and contemporary topic, the relationship of coronary artery geometry and atherosclerosis, including novel insights from CTA data analysis. There is no comprehensive and up-to-date recent review in the literature -to the best of my knowledge.

The review is  structured into 4 sections (1. Introduction 2. Technical part 3.Association of geometry with hemodynamics 4. Association of geometry with atherosclerosis), scientific sound and well written.

The topic is clinically relevant and provides basic knowledge and insights, which are relevant with regards for interpreting CTA imaging findings in practice, and correlating them with the clinical presentation and other test findings- which may even affect clinical management decisions.

Further , the authors propose a  geometric risk score”, that will be evaluated in their EU funded prospective” GEOMETRY- CTA” study protocoll ( published 2021, EHJ), a novel topic that certainly warrants systematic research. The hypothesis of the study is > that wider bifurcation angles and lower coronary artery volume index (CAVi) augmenting the atherosclerosis-prone environment and predict higher CAD extent and complexity.

Comments:

*1) p.7: section 4, line 260- 270:

The theory that arterial geometry affects atherosclerosis onset and formation is not 261 new, with reports on pulsatile flow through vascular branches and the hemodynamics of 262 atherogenesis dating back to 1975 [29]. In a study published in 1976 by Gazetopoulos et 263 al. it was shown that the length of main left coronary artery was significantly shorter in 264 patients with coronary atherosclerosis, as demonstrated on invasive angiography [30]. 265

This finding was confirmed by Saltissi et al. who showed that the LM was shorter in ath266 erosclerotic vessels, as documented on angiography. In particular, it was suggested that a short left main favored proximal atherosclerosis [31]. 268 In an early study published in 1993, Friedman et al. investigated the correlation of 269 coronary artery geometry with the distribution of early sudanophilic atherosclerotic le270 sions.

Please add a critical statement to the limitations of those 2 studies (small sample size, retrospective- bias? Were the patients matched for CVRF?, CTA not used for atherosclerosis evaluation- would be more accurate than ICA due to its 3D capability). Despite a group difference was found- which may indicate a relationship- but does not proof causality. The  LM is often variable in length, and found short but nondiseased very frequently on CTA. To my knowledge there is no data using CTA... 

*2) Line 277: *Sudanophilia? Please specific for the readers... 

ccording to google: a condition in which the leukocytes contain particles staining readily with Sudan red III.?

and line 60, “sudanophilic plaque”? please specify for the reader… vulnerable plaque? non-calcified, inflammatory?

3)line 287, when discussing and citing ref 32, Friedman, M.H., et al., Relation between coronary artery geometry and the distribution of early sudanophilic lesions. Atherosclerosis, 428 1993. 98(2): p. 193-199.

Please add a critical statement that angles were measured by ICA (and not CTA). Due to the 2D projectional nature, angle measurement are - mathematically- biased and may be not as accurate as when being measured by a 3D  imaging modality such as CTA.

4) *Please add a recent publication which used CTA  by Han et al published in JAMA 2022 and discuss findings- which are relevant for the review: (to section 4.)

Han D, Lin A, Kuronuma K et al. Association of Plaque Location and Vessel Geometry Determined by Coronary Computed Tomographic Angiography With Future Acute Coronary Syndrome-Causing Culprit Lesions. JAMA Cardiol. 2022 Mar 1;7(3):309-319. 

 the authors assessed geometry and atherosclerosis.... 

5. line 325-6: “It was found that C-shaped RCAs were 326 significantly more affected by atherosclerotic lesions in their proximal and mid regions”

> this is an interesting finding, and would speak for more atherosclerosis at more angulated parts of RCA? Can we hypothize? Level of evidence?

6. Table 1: please add a column with the number of patients included in each study.

Please try to cut text e.g.by:

Using  ICA as abbreviation for = invasive coronary angiography (in order to reduce text of the Table)

Using "CTA/CFD"  instead of (CFD analysis on CCTA).

Please try cut redundant words  e.g. significant (repetitive) = mark with asterix*, or rephrasings. 

7. line 280. RPI=? Please define the abbreviation.

8. CAVI index: For some reason, the authors did not cite the study evaluating CAVI index:

Benetos G, Benz DC, Rampidis GP et al. Coronary artery lumen volume index as a marker of flow-limiting atherosclerosis-validation against 13N-ammonia positron emission tomography. Eur Radiol. 2021 Jul;31(7):5116-5126.

which is included in the GEOMETRY study proposal . If word count allows for adding it ? I would suggest please cite and discuss study findings briefly (= correlation of small coronary volume with ischemia)

It would be valuable for the review, because its a clinically relevant observation, affecting everyday CTA interpretation.

thank you. 

Round 2

Reviewer 1 Report

The authors adequately revised the manuscript according to this reviewer's comments, and provided appropriate figures. I have no more comments about this review article.